**Data Availability Statement:** All relevant data are within the manuscript and its Supporting information files.

# Effects of aerobic exercise and resistance exercise on physical indexes and cardiovascular risk factors in obese and overweight school-age children: A systematic review and meta-analysis

**Tianhao Chen**[1⊙], **Jingxia Lin**[1⊙], **Yuzhe Lin**[1‡], **Lin Xu**[1‡], **Dian Lu**[1‡], **Fangping Li**[2‡], **Lihao Hou**[3‡], **Clare Chung Wah Yu**[1] *

**1** Department of Rehabilitation Sciences, The Hong Kong Polytechnic University, Hung Hom, Kowloon, Hong Kong, **2** Department of Biomedical Engineering, The Hong Kong Polytechnic University, Hung Hom, Kowloon, Hong Kong, **3** Department of Rehabilitation, The Second Affiliated Hospital of Nanjing University of Chinese Medicine, Nanjing, Jiangsu, China

⊙ These authors contributed equally to this work.
‡ These authors also contributed equally to this work.
* clare-chung-wah.yu@polyu.edu.hk

## Abstract

### Background

Obesity is a serious social and public health problem in the world, especially in children and adolescents. For school-age children with obesity, this stage is in the transition from childhood to adolescence, and both physical, psychological, and external environments will be full of challenges. Studies have showed that school-age children are the largest proportion of people who continue to be obese in adulthood. Physical exercise is considered as an effective way to control weight. Therefore, we focus on this point to study which factors will be improved to reduce childhood obesity.

### Objective

To assess the effects of aerobic and resistance exercise on physical indexes, such as body mass index (BMI) and body fat percentage, and cardiovascular risk factors such as VO$_2$peak, triglycerides (TG) and low-density lipoprotein (LDL), high-density lipoprotein (HDL), total cholesterol (TC), insulin and insulin resistance in school-age children who are overweight or obese.

### Method

PubMed, SPORTDiscus, Medline, Cochrane-Library, Scopus, Ovid and Web of Science were searched to locate studies published between 2000 and 2021 in obese and overweight school-age children between 6–12 years old. The articles are all randomized controlled trials (RCTs) and in English. Data were synthesized using a random-effect or a fixed-effect model

**Funding:** The author(s) received no specific funding for this work.

**Competing interests:** The authors have declared that no competing interests exist.

to analyze the effects of aerobic and resistance exercise on six elements in in school-age children with overweight or obese. The primary outcome measures were set for BMI.

## Results

A total of 13 RCTs (504 participants) were identified. Analysis of the between-group showed that aerobic and resistance exercise were effective in improving BMI (MD = -0.66; p < 0.00001), body fat percentage (MD = -1.29; p = 0.02), TG (std.MD = -1.14; p = 0.005), LDL (std.MD = -1.38; p = 0.003), TC (std.MD = -0.77; p = 0.002), $VO_2$peak (std.MD = 1.25; p = 0.001). However, aerobic and resistance exercise were not significant in improving HDL (std.MD = 0.13; p = 0.27).

## Conclusions

Aerobic exercise and resistance exercise are associated with improvement in BMI, body fat percentage, $VO_2$peak, TG, LDL, TC, while not in HDL in school-age children with obesity or overweight. Insulin and insulin resistance were not able to be analyzed in our review. However, there are only two articles related to resistance exercise in children with obesity and overweight at school age, which is far less than the number of 12 articles about aerobic exercise, so we cannot compare the effects of the two types of exercises.

## Introduction

Obesity is a serious social and public health problem in most countries in the world. In 2016, the prevalence of overweight and obesity in the global population aged 5–19 accounted for 18.5%, nearly 124 million people in this age group were obese, and 213 million people were overweight. Moreover, about 34% of children from low-income families are obese, compared with about 19% of children from high-income families [1, 2]. According to the distribution of country types, there are 35 million overweight and obese children in developing countries, 8 million in developed countries, and 92 million children globally are at risk of being overweight. The prevalence of overweight and obesity among children in Africa was 8.5% in 2010, compared with 4.9% in Asia, but the number of children affected in Asia, at 18 million, is much higher than in Africa [3]. Nowadays, childhood obesity and overweight are becoming a serious economic problem that economic losses comparable to military conflict or smoking for 2 trillion US dollars, equivalent to 2.8% of global economic output [4].

The majority of obese and overweight children and adolescents still maintain this symptom in adulthood and have adverse physical and mental health [5]. Childhood obesity is linked to elevated low-density lipoprotein (LDL), triglycerides (TG) levels, blood glucose and insulin levels and reduced high-density lipoprotein (HDL), which can easily induce cardiovascular diseases in adulthood, such as coronary heart disease, stroke, hyperlipidemia and hypertension [6, 7]. Moreover, obese and overweight people with insulin resistance are highly related to adipose tissue inflammation and prone to dysregulation of metabolic function, which increases the risk of ischemic stroke, diabetes, cancer [8, 9]. To measure overweight and obesity, Body mass index (BMI) and body fat percentage are two main indexes, and insulin resistance, triglycerides and LDL also play an essential role in measuring obesity or overweight [10–12].

It is essential to treat childhood overweight and obesity, but drugs are not allowed to treat obesity or overweight children under 12 years old due to their severe side effects [13]. In order

to prevent and treat childhood obesity or overweight, physical exercise is considered an effective way to control weight [14]. Among the various classifications of exercise, aerobic exercise are highlighted. Exercise guidelines for school-age children recommend both aerobic exercise and resistance training [15]. Peak oxygen uptake (VO2peak) is the gold standard to measure cardiopulmonary function [16]. A meta-analysis study demonstrated that aerobic training can increase heart volume and cardiac output by increasing VO2peak to reduce cardiovascular diseases [17].

There are many researches about the effect of physical activity on obesity or overweight in children and adolescents in recent 20 years [18–21]. Some long periods of follow-up studies have been proved that childhood obesity is related to sedentary behavior, and the severity of obesity is in proportion to the time of inactivity [22–24]. Children are recommended to have 1-hour physical activity every day [1]. Schwarzfischer et al. (2019) expressed that from the age of 6 to 11, the frequency of moderate to vigorous exercise remained stable until the age of 8, but decreased after the age of 11, and school-age was determined as the necessary intervention time [25]. Moreover, a study showed no significant decrease in BMI among adolescents aged 14 to 16 after 2–3 years of behavioural therapy. However, obese children participating in physical activity aged 6–8 had more effective results than obese children aged 14–16 [26].

The possibility of obese children becoming obese adults will increase with age. 26–41% of preschool children with obesity remained obese in adulthood, 42–63% of school-age children aged 6–12, and 70–80% of adolescents aged 13–17 in this trend [27, 28]. The earlier intervention in obesity, the lower the probability of obesity in adulthood. The purpose of our review was to synthesize RCTs to analyze the effects of aerobic exercise and resistance exercise on physical indexes and cardiovascular risk factors in school-age children with obesity or overweight. In addition, this is the first review to Comprehensively analyse the effects of physical exercise on physical index and cardiovascular factors in school-age children with obesity and overweight, which can have a prospective effect on future clinical and scientific research.

## Methods

### PICO and search strategy

The search keywords were operationalized using a Population, Intervention, Comparison, Outcome (PICO) chart (Table 1). The population contains school-age children with obesity or overweight from 6–12 years. There are no gender and nationality restrictions on the population.

The meta-analysis search was conducted between December 2020 and February 2021. Articles were searched from the following databases: PubMed, SPORTDiscus, Medline, Cochrane —Library, Scopus, Ovid and Web of Science, from 2000 to 2020. The main keywords used were (aerobic exercise OR resistance exercise OR physical activities OR training) AND (obesity* OR overweight OR obese*) AND (school-age children OR kid OR student) NOT adolescent. Then, this review was carried out in preferred reporting items for systematic reviews and meta-Analyses (PRISMA) flow diagram [29]. We have registered our review on the PROSPERO website with ID CRD42021241747.

**Table 1. PICO chart.**

| Population | School-age children with obesity or overweight |
|---|---|
| Intervention | Aerobic exercise and resistance exercise |
| Comparison | Other types of intervention or a placebo condition or healthy controls |
| Outcome | Physical indexes and cardiovascular risk factors |

## Study selection and eligibility criteria

All results were input to EndNote X9 to remove duplications and further assess the potential eligibility. The prespecified inclusion criteria were: 1) school-age children between 6–12 years old; 2) aerobic and resistance exercise or physical activities; 3) random control trials; 4) full-length articles in peer-reviewed journals; and 5) published in English. Studies were excluded if 1) were literature or systematic review; 2) were the pre–post-study design; 3) no BMI data; 4) had publication bias 5) had high risk of bias. Four reviewers (Tianhao CHEN, Yuzhe LIN, Dian LU and Lin XU) independently screened all articles obtained through the database. Any disagreement between reviewers was resolved by consulting Tianhao CHEN. Characteristics of included and excluded studies and data and meta-analysis are recorded by Review Manager 5.4. The extraction of information and data was recorded in the S1 and S2 Files.

## Risk of bias and publication of bias assessment

Two reviewers (Dian LU and Lin XU) independently evaluated the quality of studies using a tool for the risk of bias in randomized trials (RoB 2) from the Cochrane guideline [30]. Rob2 contains five domains to evaluate each article: sequence generation and allocation concealment (selection bias), blinding of participants and personnel (performance bias) and outcome assessors (detection bias), incomplete outcome data (attrition bias). Each domain was rated as low risk, unclear risk, or high risk. Discordances were resolved by Tianhao CHEN.

## Statistical analysis

According to Jackson & Turner (2017), meta-analysis of at least five studies is powerful, so we would not consider the analysis of the index of less than five studies [31]. Meta-analyses were conducted using the changes in the indexes between pre- and post-intervention measured. Standardized mean differences (SMDs) and mean differences (MDs) with 95% Confidence Interval (CI) were used for continuous outcomes. The MD and standard deviation (SD) were extracted by to estimate effect sizes for effects on school-age children with obesity or overweight. The standardized mean differences (SMD) were calculated using different measuring units, while the mean differences (MD) were calculated using the same measuring unit to estimate effect sizes observed in the study. Random effects models (overall effect p-value < 0.05) and fixed effects models (when overall effect p-value > 0.05 under random effects models) were two main models used to synthesize all the results in forest plots. Heterogeneity was assessed by calculating the $I^2$ statistic and $\chi 2$ test (assessing the p-value) using Review Manager V5.4 software. If the p-value was <0.1 and $I^2$ >50%, we consider the heterogeneity considerable. A random-effect model was used to combine the data if significant heterogeneity existed. In Cohen's categories, the effect size of 0.20, 0.50 and 0.80 was considered small, medium and large, respectively [32].

## Additional analyses

The risk of publication bias was observed by funnel plot at first, and then Egger regression was used to test the asymmetry of funnel plot, and p>0.1 indicated that there was no publication bias.

# Results

## Including studies

The total number of searching identified 7594 publications showed in the PRISMA flow diagram, including 2516 from PubMed, 365 from Scopus, 832 from SPORTDisscus, 1130 from

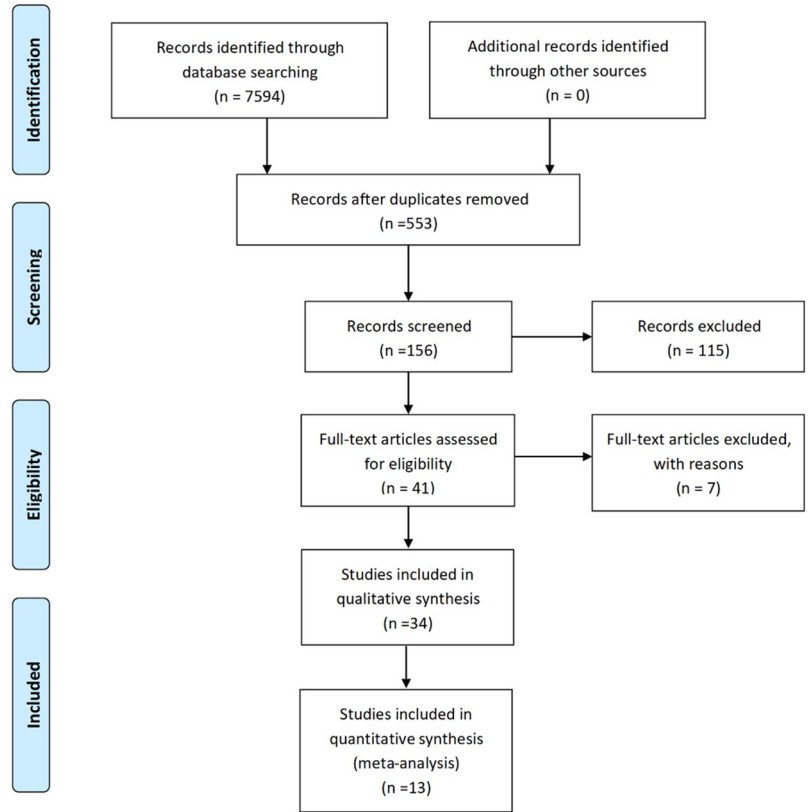

**Fig 1. PRISMA flow diagram of the studies selection process [29].**

Medline, 162 from Cochrane-Library, 1933 from Ovid and 656 from Web of Science (Fig 1). Publications screened after removing duplicates remained 553 articles. On this basis, we screened the titles and finally left 156 articles. Moreover, 156 studies were reviewed after excluding irrelevant 115 records based on abstract for two reasons: 1. the subjects were not focused on children 2. the article type was not randomized controlled trial. Our group conducted the full-text articles assessed for eligibility, and 7 studies were excluded due to no BMI data, non-English literature and no baseline or post data. 34 trials were included in the qualitative synthesis. Thirteen articles met the inclusion criteria, but 21 studies were excluded from the meta-analysis due to the average age of the subjects was not between 6 and 12 years old.

## Study information

The main messages of the 13 studies are shown in Table 2. The number of obese or overweight children in the 13 RCTs ranged from 19 to 75, and the total number is 504. 13 studies recruited participants diagnosed with obesity or overweight [33–45]. In contrast, only two studies recruited participants diagnosed with overweight only [36, 40]. BMI was assessed in all studies for physical indexes, and body fat percentage was assessed in 8 studies [33–35, 39, 41–43, 45]. In cardiovascular risk factors, $VO_2$peak was included in 8 trails [34, 35, 38–43]. TG and TC were assessed in 6 studies [34–36, 40, 43, 45], LDL, HDL were both assessed in 7 studies [34–37, 40, 45]. However, only 4 studies participants were assessed for insulin [35, 37, 40, 45], and three studies included insulin resistance (HOMR) [35, 40, 45], which cannot do meta-analysis.

**Table 2. Characteristics of the 13 studies included in the systematic review.**

| Author (year) | Participant | Sample size | Region | Mean age(sd) | Experimental group | Control group | Duration of treatment (weeks) | Details |
|---|---|---|---|---|---|---|---|---|
| Alberga et al., 2013 [33] | Obese children, BMI > 95th | 19 | Canada | 10 (2) | Resistance exercise n = 12 | Activities as usual n = 7 | 75 min/session, 2 sessions/ week, 12 weeks | Warm up: 20 min, 65–70% of max heart rate (HRmax) Resistance exercise: 45 min, 65–85% of 1-RM, 1 set, 8–12 reps; Cooldown: 10 min, light exercises and stretching. |
| Chuensiri et al., 2018 [34] | Obese children, BMI ≥2 SD above the growth reference data for boys | 48 | Thailand | 10.9 (0.93) | Aerobic exercise -HIIT = 16 Aerobic exercise-supra-HIIT = 16 | Activities as usual n = 16 | 20 min/session, 3 times/week, 12 weeks | HIIT group: eight sets of 2-min exercises at 90% of peak power output, interspersed by a 1-min rest supra-HIIT group: eight sets of 20-sec high-intensity exercises at 170% of peak power output, followed by 10 seconds of resting periods. |
| Farpour-Lambert et al., 2009 [35] | Obese children, BMI > 97th | 44 | Switzerland | 8.95 (1.5) | Aerobic exercise n = 22 | Usual care n = 22 | 60 min/day, 3 times/ week, 12 weeks | Exercise group: 30 min of aerobic exercise (fast walking, running, ball games, or swimming) at a heart rate corresponding to 55% to 65% of individual maximal cardio-respiratory fitness, followed by 20 min of strengthening exercises and 10 min of stretching and cool-down. |
| Ham et al., 2016 [36] | Overweight children, BMI > 85th | 75 | Korea | 10.77 (1.17) | Individual counselling + music skipping rope exercise n = 48 | Individual counselling + music skipping rope exercise n = 23 | 60 min/session, twice a week, 12 weeks | Experimental: 8-session individual counselling + 12-week music skipping rope exercise Control: 1-session individual counselling + 12-week music skipping rope exercise |
| Karacabey, 2009 [37] | Obese children, BMI≥30 kg/m2 | 40 | Turkey | 11.5 (0.35) | Aerobic exercise n = 20 | Usual care n = 20 | 30–65 min/day, 3 times / week, 12 weeks | Exercise group: Warm-up exercises lasting 5–10 min, 20–45 min walking–jogging exercise with a targeted heart rate reserve of 60–65%, and 5–10 min of relaxation exercises |
| McNarry et al., 2015 [38] | Obese children, BMI ≥ 95th | 26 | England | 9.3 (0.9) | Aerobic exercise n = 15 | Usual care n = 11 | 60 min/session, twice a week, 6 weeks | Exercise group: supervised high-intensity, discontinuous 6 games programme, which included ladder running, step-ups, skipping, star jumps, high knees, shuttle runs, jumping jacks, and lateral jumps. |
| Moslehi et al., 2019 [39] | Obese children, BMI 29–31 kg/m$^2$ | 30 | Iran | 10.97 (0.5) | Aerobic exercise-outdoor = 10 Aerobic exercise-indoor treadmill = 10 | Usual care n = 10 | 25–40 min/session, 3 times/ week, 8 weeks | Aerobic exercise-outdoor: football game at 65% of the reserve heart rate Aerobic exercise-indoor treadmill: treadmill at 65% of the reserve heart rate |
| Murphy et al., 2009 [40] | Overweight children, FMD response of the brachial artery <8% | 35 | America | 10.21 (1.67) | Physical activity n = 23 | Usual care n = 12 | 10–30 min/session, 5 times / week, 12 weeks | Physical activity group: Recorded daily active video game DDR use and steps they took while playing DDR |

(*Continued*)

**Table 2.** (Continued)

| Author (year) | Participant | Sample size | Region | Mean age(sd) | Experimental group | Control group | Duration of treatment (weeks) | Details |
|---|---|---|---|---|---|---|---|---|
| Prado et al., 2009 [41] | Obese children, BMI > 95th | 38 | Brazil | 10.13 (0.3) | Aerobic exercise + hypocaloric diet n = 21 | Hypocaloric diet n = 17 | 60 min/session, 3 times/ week, 16 weeks | Aerobic exercise: 30 min of walking and / or jogging on a jogging track, and 30 min of recreational exercises Dietary Protocol: Energy intake was maintained at 1800 kcal/ day. The hypocaloric diet consisted of 65% carbohydrates, 15% protein, and 20% fat |
| Prado et al., 2010 [42] | Obese children, BMI > 95th | 33 | Brazil | 10.25 (0.3) | Aerobic exercise + hypocaloric diet n = 18 | Hypocaloric diet n = 15 | 60 min/session, 3 times/ week, 16 weeks | Aerobic exercise: 30 min of walking and / or jogging on a jogging track, and 30 min of recreational exercises Dietary Protocol: Energy intake was maintained at 1800 kcal/ day. The hypocaloric diet consisted of 65% carbohydrates, 15% protein, and 20% fat |
| Saygın and Öztürk, 2011 [43] | Obese children, body fat percentage>31% | 39 | Turkey | 10–12 | Aerobic exercise n = 20 | Usual care n = 19 | 60–90 min/session, twice a week, 12 weeks | Exercise group: started with 10 to 12 min warming-up exercises and ended with 8 to 10 min cooling down exercises at an intensity of 50 to 60% of target heart rates |
| Tan et al., 2010 [44] | Chinese children body mass >20% | 60 | China | 9.45 (0.5) | Aerobic exercise n = 30 | Usual care n = 30 | 50 min/session, 5 sessions/week, 8 weeks | Exercise group: 5-min warm-up included walking, jogging, and stretching, 5- to 6-min bouts of running, jumping, squatting, crawling, and aerobic dance. 2-min of rest between exercise bouts. 5-min cool-down. |
| Zehsaz et al., 2016 [45] | Obese boy, BMI > 95th | 32 | Finland | 10.55 (0.9) | Aerobic exercise + Resistance exercise n = 16 | Usual care n = 16 | Aerobic exercise: 30–35 min/session + resistance exercise: 55 min/session, twice a week, 16 weeks | Aerobic exercise: at 55–75% HRmax, based on the Karvonen formula, as an aerobic exercise Resistance exercises: resistance training. Rubber band exercises included the squat, sit-up, seated row, knee extension, knee curl, seated leg press, overhead press, elbow curl, and bench press at 70% of the maximal single repetition. |

BMI = body mass index, FMD = flow-mediated dilation, HOMA = homeostasis model assessment, Body fat percentage = body fat percentage, HIIT = High-intensity interval trainig, DDR: Dance Dance Revolution™.

In 13 studies, aerobic exercise was applied in 12 studies [34–45], while resistance exercise was applied in 2 studies [33, 45]. Only 1 study both had the two types of exercise [45]. Walking and jogging are the most common forms of aerobic exercise in six trails [37, 40–42, 44, 45], while other experiments used football, swimming, dancing, skipping rope, and badminton. All the exercises mentioned ranged in intensity from 50–70% of the Heart Rate Reserve (HRR), representing the moderate strength in the FITT principles [46].

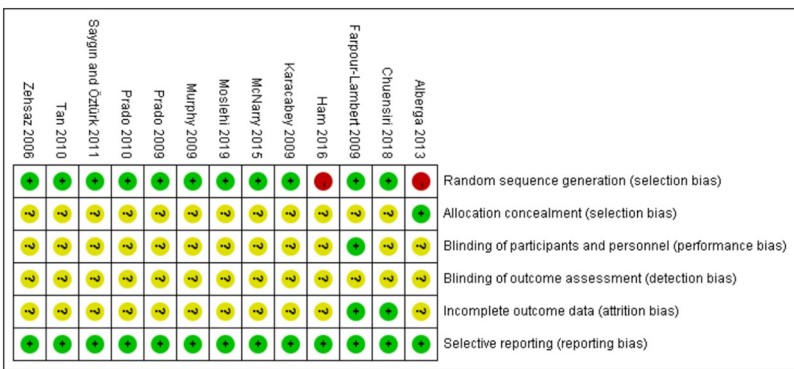

**Fig 2. Risk of bias summary.**

There were also differences in the duration of the exercise interventions, with each session ranging from 10 [40] to 90 minutes [43, 45], averaging 60 minutes per session in most studies. The trials lasted from 6 [38] to 16 weeks [40, 41, 45], with an average duration of 12 weeks.

### Risk of bias

The risk of bias was evaluated and summarized in Fig 2. Green, yellow, and red are labelled low risk, some concerns, and high risk. Thirteen studies were described as randomized, with ten studies reported detailed randomization methods, and two trials were rated as high risk [33, 36]. Only one trial reported allocation concealment in detail [33]. In detection bias, no trial described blinding of the outcome assessment in detail. Only one trial had detailed information in blinding the participants or their operators to the intervention, while other trials did not refer to the blinding information. Two studies with full participation without temporary withdrawal were regarded as low risk of attrition bias [34, 35], while the others gave no details of missing data. Moreover, all trials have a low risk of selective reporting. When all domains are rated as low risk, the overall risk can be evaluated as low risk. However, if any domain is marked as high risk, the overall risk will be high. In addition, the overall risk assessment of five domains is regarded as some concerns. Hence, only 1 study considered low risk [35], while others considered some concerns.

### Additional analyses

Egger test, and sensitivity analysis were first used in 13 studies to evaluate the publication bias on all indexes by software STATA SE16; The results showed that all indexes in obese or overweight school-age children are significant asymmetry (Egger regression P > 0.1) in the funnel plot. The "metanif" command was used for sensitivity analysis to assess the influence of a single study in meta-analysis estimation. Results remained consistent across the sensitivity analysis.

### Effect of aerobic exercise and resistance exercise on physical indexes

**Body mass index (kg/m2).**  Thirteen studies evaluated aerobic exercise and resistance exercise on BMI in obese or overweight school-age children. The total results of these 13 trials showed an overall improvement in BMI in the exercise group than in control groups (MD = -0.66; 95% CI = -0.9 to -0.43; p < 0.00001; heterogeneity: $I^2$ = 85%; $x^2$ = 94.01; p < 0.00001) (Fig 3). The studies showed an overall effect size of Z = 5.47, which represents a medium effect size. Subgroup analyses were also performed in 7 studies with different duration. The results of

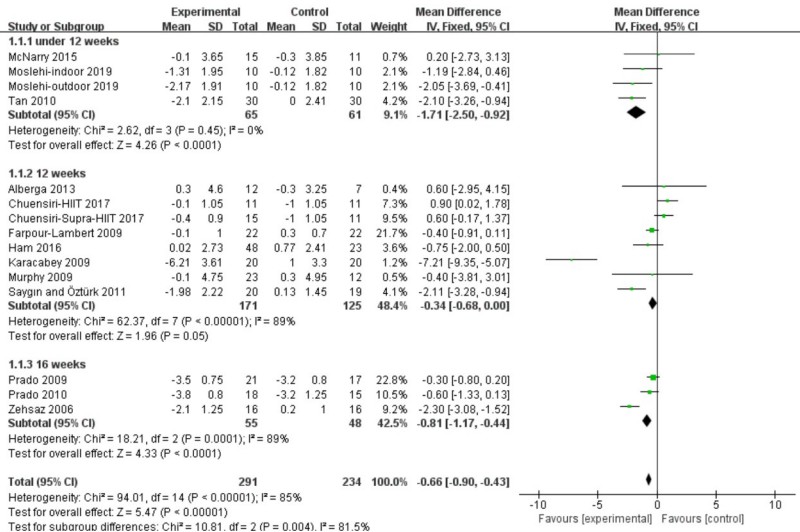

**Fig 3. Forest plot of comparison: Body mass index (kg/m2).**

7 studies with a duration of 12 weeks showed aerobic exercise and resistance exercise significantly reduced BMI level as a small effect size (MD = -0.68; 95% CI = -0.68 to 0.00; p = 0.05; heterogeneity: $I^2$ = 89%; $x^2$ = 62.37; p< 0.00001; Z = 1.96) (Fig 3).

**Body fat percentage.**   Analyses in eight studies evaluated aerobic exercise and resistance exercise on body fat percentage in obese or overweight school-age children. The total results of these 8 trials showed an overall improvement in body fat percentage in the exercise group than in control groups (MD = -1.29; 95% CI = -2.39 to -0.18; p = 0.02; heterogeneity: $I^2$ = 86%; $x^2$ = 62.72; p< 0.00001) (Fig 4). The studies showed an overall effect size of Z = 2.29, which means a small effect size.

## Effect of aerobic exercise and resistance exercise on cardiovascular risk factors

**Triglycerides.**   Analyses in six studies evaluated aerobic exercise and resistance exercise on TG in obese or overweight school-age children. The total results of these 6 trials showed an overall improvement in TG in the exercise group than in control groups (std.MD = -1.14; 95% CI = -1.93 to -0.34; p = 0.005; heterogeneity: $I^2$ = 88%; $x^2$ = 48.25; p < 0.00001) (Fig 5). The studies showed an overall effect size of Z = 2.81, which means a medium effect size.

**Low-density lipoprotein.**   Analyses in seven studies evaluated aerobic exercise and resistance exercise on LDL in obese or overweight school-age children. The total results of these 7

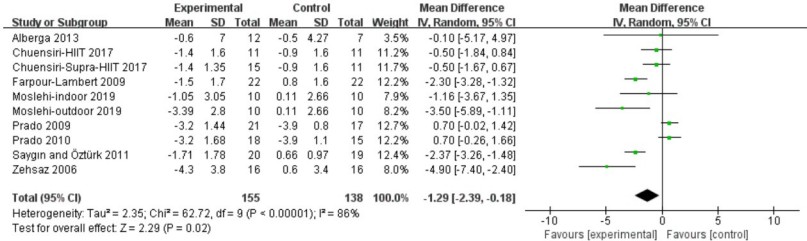

**Fig 4. Forest plot of comparison: Body fat percentage (%).**

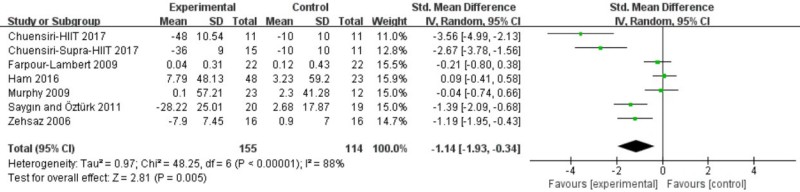

**Fig 5. Forest plot of comparison: Triglycerides.**

trials showed an overall improvement in LDL in the exercise group than in control groups (std.MD = -1.38; 95% CI = -2.57 to -0.47; p = 0.003; heterogeneity: $I^2$ = 90%; $x^2$ = 59.59; p < 0.00001) (Fig 6). The studies showed an overall effect size of Z = 2.98, which means a medium effect size.

**High-density lipoprotein.** Analyses in seven studies evaluated aerobic exercise and resistance exercise on HDL in obese or overweight school-age children. The total results of these 7 trials showed aerobic exercise and resistance exercise had no significant effects in HDL (std. MD = 0.13; 95% CI = -0.11 to 0.37; p = 0.27; heterogeneity: $I^2$ = 86%; $x^2$ = 48.79; p < 0.00001) (Fig 7).

**Total cholesterol.** Analyses in six studies evaluated aerobic exercise and resistance exercise on TC in obese or overweight school-age children. The total results of these 6 trials showed an overall improvement in TC in the exercise group than in control groups (std.MD = -0.77; 95% CI = -1.26 to -0.28; p = 0.002; heterogeneity: $I^2$ = 70%; $x^2$ = 19.93; p = 0.003) (Fig 8). The studies showed an overall effect size of Z = 3.10, which means a medium effect size.

**VO$_2$peak.** Analyses in eight studies evaluated aerobic exercise and resistance exercise on VO$_2$peak in obese or overweight school-age children. The total results of these 8 trials showed an overall improvement in VO$_2$peak in the exercise group than in control groups (std. MD = 1.25; 95% CI = 0.51 to 1.99; p = 0.001; heterogeneity: $I^2$ = 87%; $x^2$ = 70.35; p < 0.00001) (Fig 9). The studies produced a combined effect size of Z = 3.3, which represents a medium effect size.

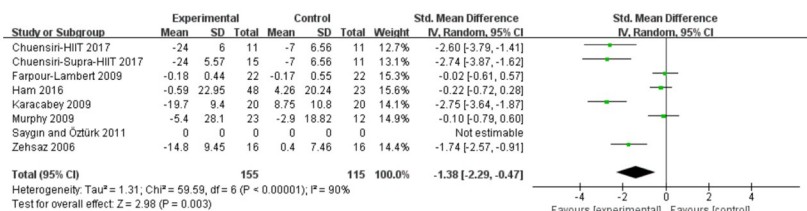

**Fig 6. Forest plot of comparison: Low-density lipoproteins.**

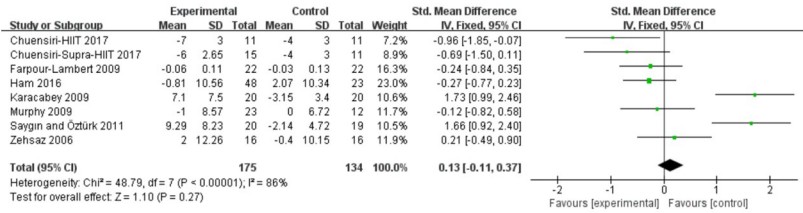

**Fig 7. Forest plot of comparison: High-density lipoproteins.**

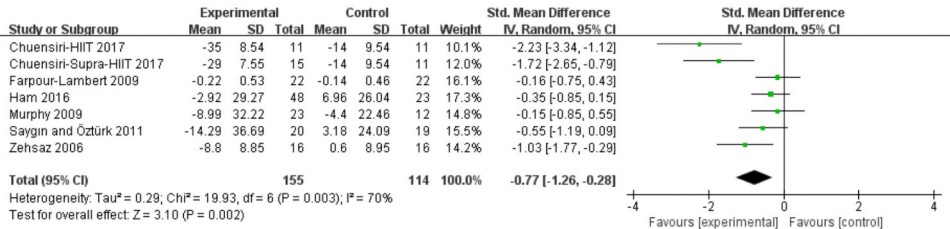

**Fig 8. Forest plot of comparison: Total cholesterol.**

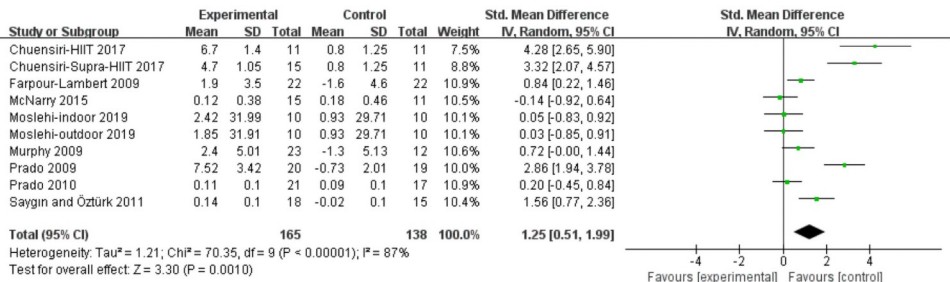

**Fig 9. Forest plot of comparison: VO$_2$peak.**

**Insulin and insulin esistance.** No more than five trials consisted of insulin [35, 37, 40, 45] and insulin resistance (HOMR) [35, 40, 45]. Due to the limited number, we did not do the meta-analysis on two indexes.

## Discussion

We carried on this systematic review and meta-analysis of 13 RCTs involving 709 school-age children with obesity or overweight. The overall data show that aerobic and resistance exercise significantly improve physical indexes and most cardiovascular risk factors in school-age children with obesity or overweight. These results demonstrated a wide range of the previous studies in aerobic exercise and resistance exercise on obesity. The data showed that the duration of the experiment varied greatly from 6 weeks to 16 weeks, and 53.85% of the studies performed physical exercise for 12 weeks. There are many kinds of sports in all the studies, among which jogging and walking are the most common methods. In general, the average exercise time was 60 minutes, and both aerobic exercise and resistance exercise was maintained at a moderate intensity of 50–70% of HRR. However, the number of trials referred to as resistance exercise in obese or overweight school-age children is much less than aerobic exercise, making it hard to compare the difference between the two types of exercise. Future studies may compare aerobic and resistance exercise effects on childhood obesity to define the specific exercise types. Some articles expressed that low HDL level is linked to obesity and overweight, but only when the threshold exercise intensities or duration is reached will improve HDL levels [6, 47, 48]. Our finding was consistent with this conclusion that aerobic and resistance exercise are not significantly associated with HDL in school-age children with obesity or overweight. These findings will be a supplement to the effect of exercise on HDL on obesity and overweight.

This study has some limitations that should be acknowledged: (1) We cannot do meta-regression to assess the relationship between the magnitude of the exercise effect and other elements. For example, only two articles referred to the diary protocol [40, 41], and there was no

information about their parents with obesity or overweight. Moreover, parents' cognition of their children's obesity will also affect the intervention of their children's obesity. Sirico and colleagues (2020) indicated that almost half of the parents misclassify their children's weight status [49]. Overweight parents were the most important risk factor for children being over-weight [50]. Diet with high fat and high-calorie habits significantly impacted childhood obesity [51, 52]. Similarly, the children's mental health and sleep factors were not taken into account. Studies found that psychological interventions can enhance and change children's motivation to exercise [53, 54]. Agaronov and colleagues (2018) found that sleep promotion is a significant element in preventing childhood obesity [55]. (2) Several articles showed that insulin and insulin resistance played an essential role in childhood obesity [56–58], but the limited number of trials tested insulin and insulin resistance (HOMR). Hence, we cannot observe the effect of exercise on insulin and insulin resistance in obese or overweight children in this age range. (3) The heterogeneity occurs in the outcome measurement procedures. (4) most trials did not report allocation concealment and blinding of participants or their operators and outcome assessment.

A large sample size cross-sectional observation study validated that more than 50% of the population mistakenly understood the real effect of exercise, and the habits and cognition of exercise in the general population are far from guidelines [59]. Both 60-minute aerobic exercise and resistance exercise are recommended safe and effective methods for preventing and treating obese or overweight school-age children. Our review and meta-analysis found that aerobic and resistance exercise positively affect physical indexes and most cardiovascular risk factors, further supporting the popularization and attention of aerobic and resistance sports in schools. However, future studies need more exercise tests on obese or overweight school-age children to provide more evidence and influence, such as insulin resistance, C-reactive protein, immunoglobin A. The number of studies on obesity or overweight of school-age children in aerobic and resistance exercise is limited. More high-quality, large sample and multi-factor further studies are required to further improve the discovery and understanding of childhood obesity in the future.

## Conclusions

Aerobic exercise and resistance exercise are associated with improvement in physical indexes, such as BMI and body fat percentage, and cardiovascular risk factors, such as TG, LDL, TC, and VO$_2$peak, while not in HDL in school-age children with obesity or overweight. This review supplements the blank and importance of exercise for obese children in school age. On the one hand, it can guide clinicians to better formulate exercise prescriptions. On the other hand, it can also provide researchers with good research direction related to school-age children with obesity and overweight and consider more factors, such as parents' heredity, eating habits and sleep quality.

## Supporting information

**S1 Checklist. PRISMA 2009 checklist.**
(DOC)

**S1 File. Extraction articles.**
(XLSX)

**S2 File. Extraction data.**
(XLSX)

## Acknowledgments

We sincerely appreciated to all authors contributed to this article and Edit age for English language editing.

## Author Contributions

**Conceptualization:** Tianhao Chen, Jingxia Lin.

**Data curation:** Tianhao Chen.

**Formal analysis:** Tianhao Chen.

**Investigation:** Yuzhe Lin, Lin Xu, Dian Lu, Fangping Li.

**Methodology:** Jingxia Lin.

**Project administration:** Clare Chung Wah Yu.

**Resources:** Jingxia Lin.

**Supervision:** Clare Chung Wah Yu.

**Validation:** Tianhao Chen.

**Writing – original draft:** Tianhao Chen, Jingxia Lin, Yuzhe Lin, Lin Xu, Dian Lu, Fangping Li.

**Writing – review & editing:** Tianhao Chen, Jingxia Lin, Yuzhe Lin, Lin Xu, Dian Lu, Fangping Li, Lihao Hou, Clare Chung Wah Yu.

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
