## [Decision Letter · Decision Letter 0]

10 Aug 2021

PONE-D-21-14627

Effects of aerobic exercise and resistance exercise on physical indexes and cardiovascular risk factors in obese and overweight school-age children: a systematic review and meta-analysis

PLOS ONE

Dear Dr. CHEN,

Thank you for submitting your manuscript to PLOS ONE. After careful consideration, we feel that it has merit but does not fully meet PLOS ONE’s publication criteria as it currently stands. Therefore, we invite you to submit a revised version of the manuscript that addresses the points raised during the review process.

We look forward to receiving your revised manuscript.

Kind regards,

Yuan-Pin Hsu

Academic Editor

PLOS ONE

Journal Requirements:

Whilst you may use any professional scientific editing service of your choice, PLOS has partnered with both American Journal Experts (AJE) and Editage to provide discounted services to PLOS authors. Both organizations have experience helping authors meet PLOS guidelines and can provide language editing, translation, manuscript formatting, and figure formatting to ensure your manuscript meets our submission guidelines. To take advantage of our partnership with AJE, visit the AJE website (http://aje.com/go/plos) for a 15% discount off AJE services. To take advantage of our partnership with Editage, visit the Editage website (www.editage.com) and enter referral code PLOSEDIT for a 15% discount off Editage services.  If the PLOS editorial team finds any language issues in text that either AJE or Editage has edited, the service provider will re-edit the text for free.

A clean copy of the edited manuscript (uploaded as the new *manuscript* file).

Reviewers' comments:

Reviewer's Responses to Questions

**Comments to the Author**

1. Is the manuscript technically sound, and do the data support the conclusions?

Reviewer #1: Yes

Reviewer #2: Partly

Reviewer #3: Yes

2. Has the statistical analysis been performed appropriately and rigorously? 

Reviewer #1: Yes

Reviewer #2: Yes

Reviewer #3: Yes

3. Have the authors made all data underlying the findings in their manuscript fully available?

Reviewer #1: Yes

Reviewer #2: Yes

Reviewer #3: Yes

4. Is the manuscript presented in an intelligible fashion and written in standard English?

Reviewer #1: Yes

Reviewer #2: No

Reviewer #3: Yes

5. Review Comments to the Author

Reviewer #1: This is a very well written paper about the role of physical exercise in obese children. Childhood obesituy is a huge public health disease and sport should be adequately prescribed in this category of subjects. This is an hot topic in literature, so it is absolutely a welcome article.

I congratulate you for your method section and for your tables and figures, very well done.

I have only minor suggestions:

- pay attention to minor grammar signs (i.e. the comma between VO2peak and triglycerides in abstract section; the arrow in Figure 1)

- in results section of abstract, try to be more coincise and not to write all those numbers in brackets: so, your results will shine

- in introduction, I suggest the following reference (https://link.springer.com/article/10.1007/s40292-019-00352-2): not only child are obese, but almost half of their parents classified their weight status incorrectly!

- in discussion, I suggest the following reference (https://pubmed.ncbi.nlm.nih.gov/32512767/), to stress the concept of the benefits of sport in several disease, such as obesity: not everyone know that sport is so good!

Reviewer #2: This systematic review aims to assess the effects of aerobic and resistance exercise on physical indexes, such as body mass index (BMI) and body fat percentage, and cardiovascular risk factors in school-age children who are overweight or obese.

The topic is very attractive and relevant; however, some issues should be addressed:

- The English language needs to be improved.

- Abstract: a short background clarifying the importance of the study question should be demonstrated.

- Conclusion: This section does not demonstrate new information. It is known that both AE and RE improve these parameters. The authors have to compare between the two types of exercise.

- Introduction: The authors should focus on the novelty and importance of the study in this section. Discuss, what is the difference between this study and other previous studies.

- Methods: How the data were analyzed in detail?

- Conclusion: In the conclusion section pleases highlight better the scientific/clinical relevance of your work. Please provide a clear “research message” of the importance of this paper in the scientific community.

Reviewer #3: This is a very interesting study about effect of exercise on chilhood obesity

Figure 2 needs to be explained more about colours and signs.

in page 14 in the first section (study selection and Eligibility criteria) last sentence is incomplete.

6. PLOS authors have the option to publish the peer review history of their article (what does this mean?). If published, this will include your full peer review and any attached files.

Reviewer #1: No

Reviewer #2: **Yes: **Walid Kamal Abdelbasset

Reviewer #3: No

---

## [Author Response · Author response to Decision Letter 0]

22 Aug 2021

Dear Mr Hsu and reviewers

Thank you for your letter and the reviewers’ comments on our manuscript entitled " Effects of aerobic exercise and resistance exercise on physical indexes and cardiovascular risk factors in obese and overweight school-age children: a systematic review and meta-analysis " [EMID: a082f3b94beff237]. Those comments are very helpful for revising and improving our paper, as well as the important guiding significance to another research. We have studied the comments carefully and made corrections which we hope to meet with approval. The main corrections are in the manuscript and the responds to the reviewers’ comments are as follows (the replies are highlighted in blue).

Kind regards.

Tianhao CHEN

E-mail: 20076293g@connect.polyu.hk

Corresponding author: Clare Chung Wah YU

E-mail address: clare-chung-wah.yu@polyu.edu.hk

---

## [Decision Letter · Decision Letter 1]

25 Aug 2021

Effects of aerobic exercise and resistance exercise on physical indexes and cardiovascular risk factors in obese and overweight school-age children: a systematic review and meta-analysis

PONE-D-21-14627R1

Dear Dr. CHEN,

We’re pleased to inform you that your manuscript has been judged scientifically suitable for publication and will be formally accepted for publication once it meets all outstanding technical requirements.

Kind regards,

Yuan-Pin Hsu

Academic Editor

PLOS ONE

Additional Editor Comments (optional):

Reviewers' comments:

Reviewer's Responses to Questions

**Comments to the Author**

1. If the authors have adequately addressed your comments raised in a previous round of review and you feel that this manuscript is now acceptable for publication, you may indicate that here to bypass the “Comments to the Author” section, enter your conflict of interest statement in the “Confidential to Editor” section, and submit your "Accept" recommendation.

Reviewer #1: All comments have been addressed

Reviewer #2: All comments have been addressed

2. Is the manuscript technically sound, and do the data support the conclusions?

Reviewer #1: Yes

Reviewer #2: Yes

3. Has the statistical analysis been performed appropriately and rigorously? 

Reviewer #1: Yes

Reviewer #2: Yes

4. Have the authors made all data underlying the findings in their manuscript fully available?

Reviewer #1: Yes

Reviewer #2: Yes

5. Is the manuscript presented in an intelligible fashion and written in standard English?

Reviewer #1: Yes

Reviewer #2: Yes

6. Review Comments to the Author

Reviewer #1: Thanks for you revisions and your answer!

Now the manuscript has improved and I think is ready to be published.

Reviewer #2: All elements of the study are well-presented. No additional revisions are required. I suggest accept this manuscript for publication in the current version.

7. PLOS authors have the option to publish the peer review history of their article (what does this mean?). If published, this will include your full peer review and any attached files.

Reviewer #1: No

Reviewer #2: **Yes: **Walid Kamal Abdelbasset

---

## [Editor Report · Acceptance letter]

27 Aug 2021

PONE-D-21-14627R1 

Effects of aerobic exercise and resistance exercise on physical indexes and cardiovascular risk factors in obese and overweight school-age children: a systematic review and meta-analysis 

Dear Dr. Chen:

I'm pleased to inform you that your manuscript has been deemed suitable for publication in PLOS ONE. Congratulations! Your manuscript is now with our production department. 

Kind regards, 

on behalf of

Dr. Yuan-Pin Hsu 

Academic Editor

PLOS ONE